# Manufacturing and Characterization of Tube-Filled ZA27 Metal Foam Heat Exchangers

**Thomas Fiedler** *[ID], **Ryan Moore** and **Nima Movahedi** [ID]

Centre for Mass and Thermal Transport, College of Engineering, Science and Environment, Callaghan Campus, The University of Newcastle, Callaghan, NSW 2308, Australia; Ryan.W.Moore@uon.edu.au (R.M.); nima.movahedi@uon.edu.au (N.M.)
* Correspondence: Thomas.Fiedler@newcastle.edu.au

**Abstract:** This study investigates the heat transfer performance of a novel ZA27 metal foam heat exchanger. An open-celled metal foam is combined with a thin-walled copper tube in a single-step casting process. The heat transfer between two separated water streams flowing through the copper tube and foam, respectively, is measured and compared to an equivalent shell tube heat exchanger arrangement. Heat transfer enhancement of up to 71% and a heat transfer rate exceeding 30 kW are observed and attributed to the increased surface area of the metallic foam. However, overall performance was limited by the inefficient heat transfer between the internal mass stream and the copper tube.

**Keywords:** heat exchanger; tube-filled metal foam; heat transfer

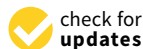



## 1. Introduction

Metallic foams combine a metallic matrix with controlled porosity. Closed-cell metallic foams exhibit superior mechanical properties and are predominantly considered for structural applications [1]. Open-cell metallic foams combine a highly conductive metallic matrix with a large volumetric surface area. This combination suggests their use for functional applications such as heat exchangers [2], electrodes [3] or sound absorbers [4].

The current study addresses the integration of ZA27 metal foams into compact heat exchangers to improve heat transfer. Zhao et al. [5] conducted a numerical investigation of open-cell foam-filled-tube heat exchangers. According to their results, increased pore-density enhances the heat transfer capacity of this heat exchanger. In addition, they observed that metal foam heat exchangers outperformed the finned tube heat exchangers that were considered in this study. This was explained by the higher surface area of the metal foam and the thorough mixing of flow in its porous structure. T'Joen et al. [6] studied the performance of a heat exchanger formed of an aluminium tube wrapped with thin layers (4–8 mm) of aluminium foam inside a wind tunnel. Their results showed the importance of achieving a low thermal contact resistance between tube and foam, e.g., by using a brazing process. They further investigated the influence of the foam height on the performance of the heat exchanger setup. The results showed that increasing the foam height reduces the exterior convective heat transfer resistance, but simultaneously increases the pressure drop. In another study [7], metal foam-wrapped tubes in a shell and tube condenser were examined under sloshing conditions. To this end, open-cell copper foams with pore densities of 5, 10, 20 and 40 pores per inch (ppi) were joined to a copper tube using a brazing process. Foam tubes paired with 10 ppi copper foam showed the highest condensation heat transfer capacity. Increasing the pore density to more than 20 ppi resulted in a similar condensation heat transfer performance as a plain tube. In [8], a thin (4 mm diameter) copper tube was filled with 20 and 30 ppi copper foams. The foam-filled tubes were experimentally investigated in a heat exchanger setup using a refrigerant fluid. The pressure drop and heat transfer coefficient showed a strong dependency on

the metal foam's geometry. Most of the predictive models failed to accurately predict the experimental data. This was attributed to imperfections in the metal foams introduced when small foam cylinders were cut from a metal foam sheet. Broken ligaments and imperfect pores in the cross-sectional area decreased the pressure drop and heat transfer coefficient. A numerical study [9] compared the heat transfer capacity of aluminium and copper foams in an exhaust gas heat-exchanger system. The heat transfer rate of the copper tube for a fluid velocity of 30 m/s was 4–9% higher compared to the aluminium foam. However, given the higher cost of copper foam, usage of the aluminium foam was recommended for this specific application. In [10], a novel shell and tube heat exchanger containing metal foam was used to recover heat from exhaust gas. A 3D numerical model was used to study the thermal-hydraulic performance of metallic foam baffle heat exchanger systems. In comparison with conventional baffle heat exchangers, the application of metal foams considerably improved waste heat recovery. It was observed that increasing the metal foam's thickness enhanced system performance. In [11], regular metallic porous lattices were considered for air-side heat transfer enhancement in heat exchangers. Lattice structures made of $AlSi_{10}Mg$ alloy were manufactured using the selective laser-melting (SLM) technique. The lattice structures were designed by a repetition of Rhombi-Octet unit cells. Two different unit cell sizes of 7 and 14 mm were considered to fabricate full-sized lattice heat exchangers. Experimental results showed that decreasing the unit cell size improved thermal conductance and the heat transfer coefficient of the porous lattice heat exchangers by 40–45%. However, the pressure drop across the metallic lattice with a smaller unit cell size was higher. A comparison with conventional fin-tube heat exchangers showed that lattice heat exchangers have a significantly higher heat transfer coefficient, likely due to the formation of eddies downstream of ligaments that increase fluid mixing. The performance of a heat sink using phase change material (PCM) embedded in metal foams was numerically and experimentally investigated in [12]. To this end, copper and nickel open-cell foams with different pore densities were infiltrated with paraffin wax as the PCM. Compared to pure paraffin, the heat diffusion within the composite materials was improved by a factor of 2–4. Heat diffusion through the composite was inversely proportional to foam porosity. It was further shown that the insertion of the metal foams slightly decreased the storage capacity of the heat sink. The influence of pore density on the heat transfer and pressure drop in moist air flowing through hydrophilic metal foams was studied in [13]. The results showed that both the heat transfer coefficient and the pressure drop increased with the pore density. In copper foams with a hydrophilic coating, the overall heat transfer coefficient was improved by 2–21%. In addition, at velocities above 0.75 m/s, the pressure drop of wet air in the coated metal foams was reported to be 1–15% smaller compared to their uncoated counterparts.

The main novelty of this study is the usage of an integrated casting technique that enables a close bond between the internal heat transfer tube and the surrounding foam. As a result, thermal contact resistance between these components is minimized, which is expected to improve the heat transfer performance of the heat exchanger. The second novelty of the current study is the usage of a low-cost metallic foam obtained by leaching salt particles from a ZA27 alloy matrix. Compact foam heat exchangers are tested in a counter-flow arrangement and their heat transfer performance is quantified for various mass flow rates.

## 2. Materials and Methods

Tube-filled open-cell metal foams were manufactured using the counter-gravity infiltration casting technique described in [14]. A schematic of the manufacturing setup is shown in Figure 1a. A double-ended metal tube was inserted and centred inside a graphite mould. The volume between the metal tube and mould was then filled with spherical NaCl particles with diameters of 2–2.5 mm. Particle size was controlled by sieving the received particles with 2.5 mm and 2 mm meshed. The mould was filled in several batches, interrupted by tapping and vibration, to minimize any gradients in their packing density.

A piece of stainless-steel mesh was inserted at the open end of the mould to keep the metal tube and particles in place. Next, a piece of ZA27 alloy was placed in a graphite crucible. The chemical composition of the ZA27 alloy obtained from Hayes Metals Pty Ltd., Riverstone, NSW, Australia [15] is shown in Table 1.

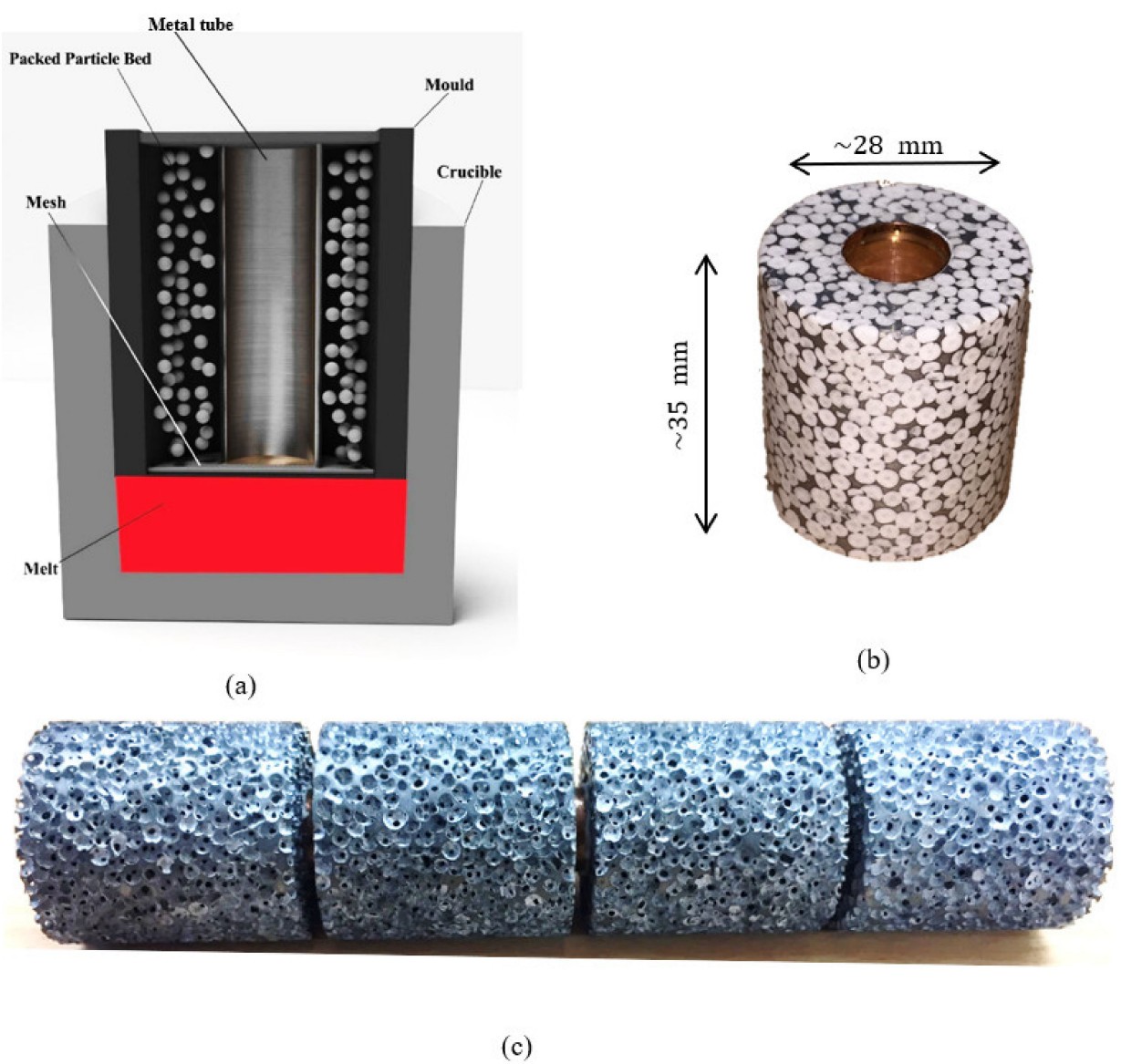

**Figure 1.** (**a**) Manufacturing setup, (**b**) sample before leaching and (**c**) foam heat exchanger module.

**Table 1.** Chemical composition of ZA27 alloy used to manufacture metal foams [15].

| Element | Al | Cu | Mg | Fe | Pb | Cd | Zn |
|---------|-----|-------|-----|-------|--------|---------|---------|
| Wt% | $25-28$ | $2-2.5$ | 1 | 0.075 | $<0.06$ | $<0.006$ | Balance |

The mould was then rotated upside down and inserted into the same crucible. The assembly was placed in an electric furnace and heated to 535 °C in an Argon atmosphere for 30 min. After melting the ZA27 alloy, the assembly was removed from the furnace and casting was completed by placing a 2 kg weight on top of the mould. Following cooling at atmospheric conditions, the solidified sample was removed from the mould. Both ends of a sample were machined to obtain surfaces with open porosity. Figure 1b shows a tube-filled composite sample before leaching the NaCl particles. Particle leaching was performed

by immersing the sample in warm water (70 °C) for one hour. Four individual samples with similar characteristics were manufactured and stacked to assemble the compact heat exchanger setup shown in Figure 1c. ZA27 foams exhibit a macroscopic density of 1.28–1.36 g/cm$^3$ and an interconnected porosity of 72–74 vol.% [15]. To contain the external mass flow through the foam, the assembly was enclosed by a plexiglass tube. For comparison, a single tube shell heat exchanger was tested, where the copper tube was identical to the tube in the tube-filled foam samples.

## 2.1. Experimental Setup

A schematic of the experimental setup is shown in Figure 2. A modified Armfield HT30XC shell and tube heat exchanger setup was used to perform the tests. Cool water was extracted directly from a tap and directed through the outer tube. This arrangement was selected to minimize stray heat transfer losses due to the lower temperature difference between tap water and ambient air. After passing through the assembly, the heated tap water was discharged into a sink. The second mass stream formed a closed loop. Water was heated to the target temperature within a reservoir (i.e., the hot water heater) and pumped through the inner tube of the heat exchanger before reheating. The two mass flow rates were regulated using two RS 257–149 flow meters. All temperatures were measured using RS Pro IEC type K thermocouples with an accuracy of ±1.5% within the considered temperature range.

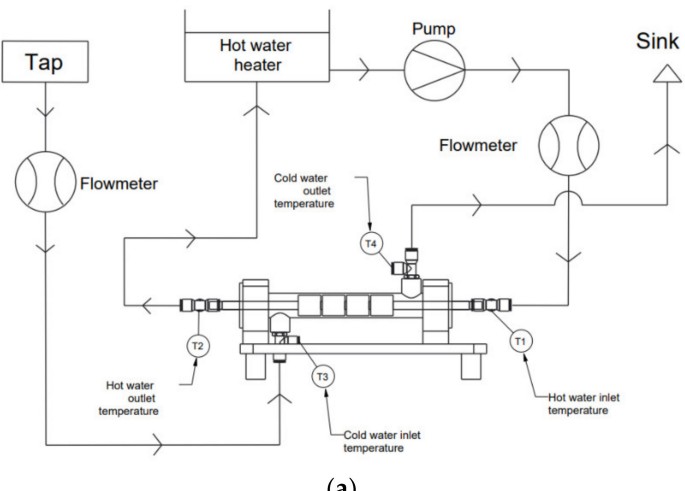

(**a**)

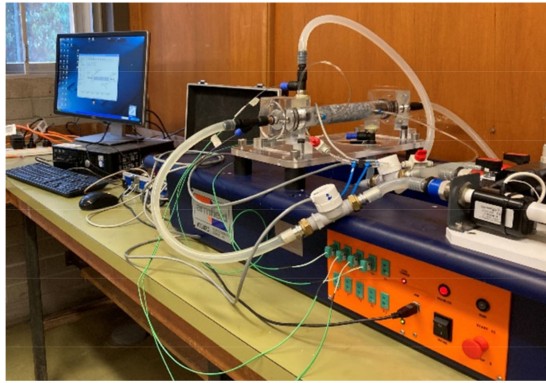

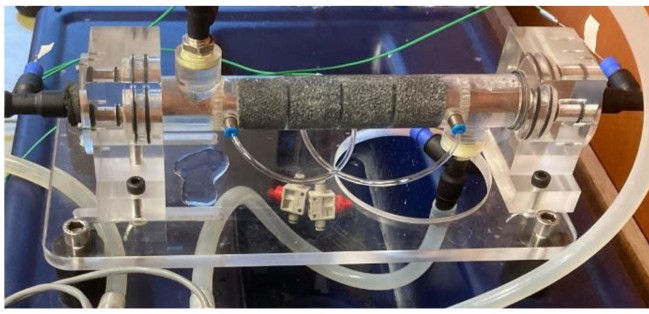

(**b**)

**Figure 2.** (**a**) Schematic of the experimental setup. (**b**) Heat exchanger assembly used for experimental study.

### 2.2. Thermodynamic Analysis

Figure 3 shows the thermodynamic model of the heat exchanger. All measurements were conducted at steady-state; therefore, the system properties mass $m$, Energy $E$, local temperature $T_i$ and local enthalpy $h_i$ are time-independent. The following two mass streams were controlled using flow meters: $\dot{m}_{int}$ flowing through the internal hollow tube and $\dot{m}_{ext}$ directed through the external tube containing the metallic foam. Temperature sensors at the inlets and outlets allowed for measurement of the four temperatures $T_i$. Temperature differences between the mass streams trigger the internal heat transfer $\dot{Q}_{int}$. For efficient heat exchange, this value must be maximised. In contrast, the stray heat transfer $\dot{Q}_{ext}$ is often considered a loss term, as this thermal energy is dissipated to the surroundings.

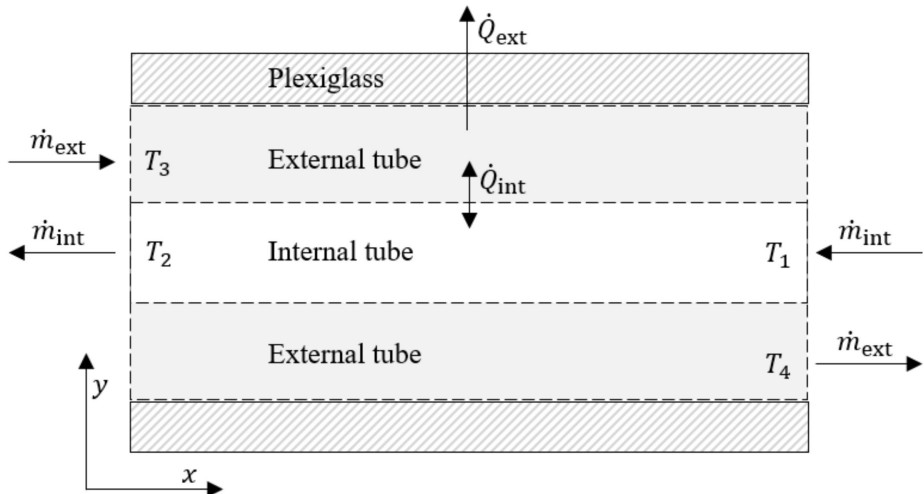

**Figure 3.** Thermodynamic model.

A first law analysis of the internal tube yields

$$\frac{dE}{dt} = \dot{Q}_{int} - \dot{W} + \dot{m}_{int} \cdot \left( h_{int,2} - h_{int,1} + \frac{1}{2}\left( v_{int,2}^2 - v_{int,1}^2 \right) + g(y_{int,2} - y_{int,1}) \right) \tag{1}$$

where $v$ represents the local velocity and $y$ the elevation. For steady-state analysis, the change of system energy is zero, i.e., $\frac{dE}{dt} = 0$. Furthermore, no power $\dot{W}$ is transferred and changes in the kinetic and gravitational potential energies are negligible. These assumptions permit the simplification of Equation (1) to

$$0 = \dot{Q}_{int} + \dot{m}_{int} \cdot (h_{int,2} - h_{int,1}) \tag{2}$$

Water entering and leaving the heat exchanger is in the subcooled (compressed) liquid state. Enthalpies can, therefore, be approximated using saturated liquid (SL) data at the same temperature $T$, i.e., $h(T,p) \approx h_{SL}(T)$. The MATLAB library XSteam [16] was utilized for the determination of all enthalpies. By solving this, we obtain, for the internal heat transfer,

$$\dot{Q}_{int} = \dot{m}_{int} \cdot (h_{SL}(T_1) - h_{SL}(T_2)) \tag{3}$$

Analogously, we can derive for the outer tube

$$\dot{Q}_{int} = \dot{m}_{ext} \cdot (h_{SL}(T_4) - h_{SL}(T_3)) - \dot{Q}_{ext} \tag{4}$$

For the steady-state, and based on the sensor data accuracy of $\pm$ 1.5% for the RS 257–149 flow meters and $\pm$ 1.5% K for the RS Pro IEC thermocouples, the analysis of experimental uncertainty [17] yields $\pm$ 3.35% for the internal heat transfer $\dot{Q}_{int}$.

## 3. Results and Discussion

All measurements were conducted in the counterflow configuration shown in Figure 3. Six different combinations of the internal and external mass flow rates $\dot{m}_{int}$, $\dot{m}_{ext}$ were considered, and three datasets were obtained per configuration to ensure the repeatability of measurements. The measurement data of the tube-filled foam heat exchangers and the calculated internal heat transfer rates are shown in Table 2. In general, the heat transfer $\dot{Q}_{int}$ from the warm to the cool mass stream increases with mass flow rates $\dot{m}_{int}$ and $\dot{m}_{ext}$. Comparing the mass flow rate configurations #1 and #2, a doubling of the external mass flow rate $\dot{m}_{ext}$ (whilst keeping $\dot{m}_{int}$ approximately constant) results in only a minor increase in $\dot{Q}_{int}$, from an average of 18.33 kW to 21.07 kW. In contrast, increasing the internal mass flow rate $\dot{m}_{int}$ from 1.91 kg/s (configuration #4) to 2.96 kg/s (configuration #5), whilst keeping $\dot{m}_{ext}$ unchanged, results in a distinct increase in $\dot{Q}_{int}$ from 35.08 kW to 43.91 kW. It can be concluded that the rate of heat transfer shows higher sensitivity to a variation in the internal mass flow through the hollow tube.

**Table 2.** Experimental measurement data of tube-filled foam heat exchangers.

| | $\dot{m}_{ext}$ | $\dot{m}_{int}$ | $T_1$ | $T_2$ | $T_3$ | $T_4$ | $\dot{Q}_{int}$ |
|---|---|---|---|---|---|---|---|
| | [kg/s] | [kg/s] | [°C] | [°C] | [°C] | [°C] | [kW] |
| #1 | 0.93 $\pm$ 0.01 | 0.91 $\pm$ 0.01 | 59.48 $\pm$ 0.89 | 54.71 $\pm$ 0.89 | 18.70 $\pm$ 0.28 | 23.79 $\pm$ 0.36 | 18.16 $\pm$ 0.61 |
| | 0.91 $\pm$ 0.01 | 0.90 $\pm$ 0.01 | 59.89 $\pm$ 0.90 | 55.18 $\pm$ 0.83 | 18.83 $\pm$ 0.28 | 24.07 $\pm$ 0.36 | 17.93 $\pm$ 0.60 |
| | 0.90 $\pm$ 0.01 | 0.92 $\pm$ 0.01 | 60.11 $\pm$ 0.90 | 55.20 $\pm$ 0.83 | 18.58 $\pm$ 0.28 | 23.96 $\pm$ 0.36 | 18.89 $\pm$ 0.63 |
| #2 | 1.91 $\pm$ 0.03 | 0.92 $\pm$ 0.01 | 59.98 $\pm$ 0.90 | 54.44 $\pm$ 0.82 | 18.52 $\pm$ 0.28 | 21.56 $\pm$ 0.32 | 21.32 $\pm$ 0.71 |
| | 1.90 $\pm$ 0.03 | 0.89 $\pm$ 0.01 | 60.20 $\pm$ 0.90 | 54.63 $\pm$ 0.82 | 18.49 $\pm$ 0.28 | 21.47 $\pm$ 0.32 | 20.73 $\pm$ 0.69 |
| | 1.92 $\pm$ 0.03 | 0.88 $\pm$ 0.01 | 60.40 $\pm$ 0.91 | 54.65 $\pm$ 0.82 | 18.32 $\pm$ 0.27 | 21.33 $\pm$ 0.32 | 21.16 $\pm$ 0.71 |
| #3 | 0.90 $\pm$ 0.01 | 2.92 $\pm$ 0.04 | 60.01 $\pm$ 0.90 | 57.08 $\pm$ 0.86 | 18.57 $\pm$ 0.28 | 28.68 $\pm$ 0.43 | 35.79 $\pm$ 1.20 |
| | 0.92 $\pm$ 0.01 | 2.90 $\pm$ 0.06 | 60.00 $\pm$ 0.90 | 57.07 $\pm$ 0.86 | 18.67 $\pm$ 0.28 | 28.59 $\pm$ 0.43 | 35.54 $\pm$ 1.19 |
| | 0.93 $\pm$ 0.01 | 2.85 $\pm$ 0.04 | 60.07 $\pm$ 0.90 | 57.12 $\pm$ 0.86 | 18.65 $\pm$ 0.28 | 28.50 $\pm$ 0.43 | 35.29 $\pm$ 1.18 |
| #4 | 1.93 $\pm$ 0.03 | 1.92 $\pm$ 0.03 | 60.05 $\pm$ 0.90 | 55.67 $\pm$ 0.84 | 18.61 $\pm$ 0.28 | 23.39 $\pm$ 0.35 | 35.18 $\pm$ 1.18 |
| | 1.91 $\pm$ 0.03 | 1.92 $\pm$ 0.03 | 60.10 $\pm$ 0.90 | 55.72 $\pm$ 0.84 | 18.65 $\pm$ 0.28 | 23.45 $\pm$ 0.35 | 35.18 $\pm$ 1.18 |
| | 1.91 $\pm$ 0.03 | 1.90 $\pm$ 0.03 | 59.96 $\pm$ 0.90 | 55.57 $\pm$ 0.83 | 18.59 $\pm$ 0.28 | 23.36 $\pm$ 0.35 | 34.89 $\pm$ 1.17 |
| #5 | 1.88 $\pm$ 0.03 | 2.94 $\pm$ 0.04 | 60.01 $\pm$ 0.90 | 56.45 $\pm$ 0.85 | 18.44 $\pm$ 0.28 | 24.51 $\pm$ 0.37 | 43.78 $\pm$ 1.47 |
| | 1.86 $\pm$ 0.03 | 2.98 $\pm$ 0.04 | 60.05 $\pm$ 0.90 | 56.50 $\pm$ 0.85 | 18.47 $\pm$ 0.28 | 24.62 $\pm$ 0.37 | 44.25 $\pm$ 1.48 |
| | 1.85 $\pm$ 0.3 | 2.96 $\pm$ 0.04 | 59.93 $\pm$ 0.90 | 56.40 $\pm$ 0.85 | 17.61 $\pm$ 0.26 | 23.75 $\pm$ 0.36 | 43.71 $\pm$ 1.46 |
| #6 | 2.85 $\pm$ 0.04 | 2.91 $\pm$ 0.04 | 60.02 $\pm$ 0.90 | 56.00 $\pm$ 0.84 | 17.84 $\pm$ 0.27 | 22.25 $\pm$ 0.33 | 48.93 $\pm$ 1.64 |
| | 2.85 $\pm$ 0.04 | 2.95 $\pm$ 0.04 | 59.95 $\pm$ 0.90 | 56.00 $\pm$ 0.84 | 18.06 $\pm$ 0.27 | 22.52 $\pm$ 0.34 | 48.74 $\pm$ 1.63 |
| | 2.86 $\pm$ 0.04 | 2.95 $\pm$ 0.04 | 59.98 $\pm$ 0.90 | 56.03 $\pm$ 0.84 | 18.05 $\pm$ 0.27 | 22.53 $\pm$ 0.34 | 48.74 $\pm$ 1.63 |

As a reference, all experiments were repeated using a single tube-shell heat exchanger arrangement. Apart from the ZA27 foam being removed, this setup is identical to the foam-filled tube heat exchanger. To this end, the foam heat exchanger (see Figure 2) was replaced by a single, straight copper tube cut from the same tube that was used for the manufacturing of the foam elements. The obtained measurement data are summarized in Table 3 below.

**Table 3.** Experimental measurement data of tube heat exchangers.

|  | $\dot{m}_{ext}$ | $\dot{m}_{int}$ | $T_1$ | $T_2$ | $T_3$ | $T_4$ | $\dot{Q}_{int}$ |
|---|---|---|---|---|---|---|---|
|  | [kg/s] | [kg/s] | [°C] | [°C] | [°C] | [°C] | [kW] |
| #1 | $0.94 \pm 0.01$ | $0.95 \pm 0.01$ | $60.30 \pm 0.90$ | $56.11 \pm 0.84$ | $16.09 \pm 0.24$ | $20.53 \pm 0.31$ | $16.65 \pm 0.56$ |
|  | $0.92 \pm 0.01$ | $0.98 \pm 0.01$ | $60.04 \pm 0.90$ | $56.18 \pm 0.84$ | $15.86 \pm 0.24$ | $19.94 \pm 0.30$ | $15.82 \pm 0.53$ |
|  | $0.93 \pm 0.01$ | $0.96 \pm 0.01$ | $59.99 \pm 0.90$ | $56.29 \pm 0.84$ | $15.81 \pm 0.24$ | $19.51 \pm 0.29$ | $14.39 \pm 0.48$ |
| #2 | $1.91 \pm 0.03$ | $0.96 \pm 0.01$ | $60.12 \pm 0.90$ | $55.48 \pm 0.83$ | $13.88 \pm 0.21$ | $16.14 \pm 0.24$ | $18.63 \pm 0.62$ |
|  | $1.91 \pm 0.03$ | $0.96 \pm 0.01$ | $60.03 \pm 0.90$ | $55.44 \pm 0.83$ | $14.14 \pm 0.21$ | $16.40 \pm 0.25$ | $18.43 \pm 0.62$ |
|  | $1.90 \pm 0.03$ | $0.98 \pm 0.01$ | $59.96 \pm 0.90$ | $55.40 \pm 0.83$ | $14.63 \pm 0.22$ | $16.90 \pm 0.25$ | $18.69 \pm 0.63$ |
| #3 | $0.94 \pm 0.01$ | $2.84 \pm 0.04$ | $60.62 \pm 0.91$ | $58.78 \pm 0.88$ | $15.79 \pm 0.24$ | $21.53 \pm 0.32$ | $21.86 \pm 0.73$ |
|  | $0.96 \pm 0.01$ | $2.84 \pm 0.03$ | $60.00 \pm 0.90$ | $58.20 \pm 0.87$ | $15.81 \pm 0.24$ | $21.39 \pm 0.33$ | $21.39 \pm 0.72$ |
|  | $0.96 \pm 0.01$ | $2.84 \pm 0.04$ | $59.98 \pm 0.90$ | $58.14 \pm 0.87$ | $15.87 \pm 0.24$ | $21.61 \pm 0.32$ | $21.86 \pm 0.73$ |
| #4 | $1.91 \pm 0.03$ | $1.90 \pm 0.03$ | $60.28 \pm 0.90$ | $57.07 \pm 0.86$ | $15.21 \pm 0.23$ | $18.39 \pm 0.28$ | $25.51 \pm 0.85$ |
|  | $1.89 \pm 0.03$ | $1.88 \pm 0.03$ | $60.00 \pm 0.90$ | $56.82 \pm 0.85$ | $15.27 \pm 0.23$ | $18.42 \pm 0.28$ | $25.01 \pm 0.84$ |
|  | $1.88 \pm 0.03$ | $1.89 \pm 0.03$ | $59.98 \pm 0.90$ | $56.82 \pm 0.85$ | $15.26 \pm 0.23$ | $18.46 \pm 0.28$ | $24.98 \pm 0.84$ |
| #5 | $1.90 \pm 0.03$ | $2.81 \pm 0.04$ | $60.56 \pm 0.91$ | $58.36 \pm 0.88$ | $15.32 \pm 0.23$ | $19.04 \pm 0.29$ | $25.86 \pm 0.87$ |
|  | $1.89 \pm 0.03$ | $2.83 \pm 0.04$ | $60.04 \pm 0.90$ | $57.89 \pm 0.87$ | $15.33 \pm 0.23$ | $18.83 \pm 0.28$ | $25.45 \pm 0.85$ |
|  | $1.90 \pm 0.03$ | $2.89 \pm 0.04$ | $59.98 \pm 0.90$ | $57.86 \pm 0.87$ | $15.39 \pm 0.23$ | $18.95 \pm 0.28$ | $25.63 \pm 0.86$ |
| #6 | $2.81 \pm 0.04$ | $2.80 \pm 0.04$ | $60.66 \pm 0.91$ | $58.05 \pm 0.87$ | $15.33 \pm 0.23$ | $18.24 \pm 0.27$ | $30.57 \pm 1.02$ |
|  | $2.84 \pm 0.04$ | $2.83 \pm 0.04$ | $60.03 \pm 0.90$ | $57.44 \pm 0.86$ | $15.29 \pm 0.23$ | $18.12 \pm 0.27$ | $30.66 \pm 1.03$ |
|  | $2.84 \pm 0.04$ | $2.84 \pm 0.04$ | $60.00 \pm 0.90$ | $57.39 \pm 0.86$ | $15.30 \pm 0.23$ | $18.23 \pm 0.27$ | $31.01 \pm 1.04$ |

Figure 4 shows the heat transfer $\dot{Q}_{int}$ plotted against the internal mass flow rate $\dot{m}_{int}$ for the tube-filled foam (filled markers) and the single tube (empty markers) heat exchangers. As expected, tube-filled foam heat exchangers achieve higher heat transfer rates for all configurations. In addition, with increasing internal mass flow rate a distinct increase in the heat transfer is observed, particularly for the tube-filled foam heat exchanger. The heat transfer $\dot{Q}_{int}$ can be abstracted as a scale, where the heat addition to the cold mass stream must balance the heat rejection of the warm mass stream. It appears that the efficient heat transfer between the foam and the cold mass stream results in a similar temperature of liquid and foam material. As a result, increasing the external mass flow rate $\dot{m}_{ext}$ does not significantly increase the heat transfer $\dot{Q}_{int}$. Conversely, a less efficient heat transfer between the internal mass stream $\dot{m}_{int}$ and the hollow copper tube (due to a much smaller contact area) likely results in a larger temperature gradient. Heat transfer can thus be improved by increasing the internal mass flow rate $\dot{m}_{int}$ until this temperature gradient eventually diminishes. For the same reason, the heat transfer of the single tube heat exchanger is less sensitive to the internal mass flow rate, as its surface areas that are in contact with the internal and external mass streams are similar. For example, for the external mass flow rate $\dot{m}_{ext} \approx 1.9$ kg/s, the heat transfer is nearly identical for $\dot{m}_{int} \approx 1.9$ kg/s and $\dot{m}_{int} \approx 2.9$ kg/s.

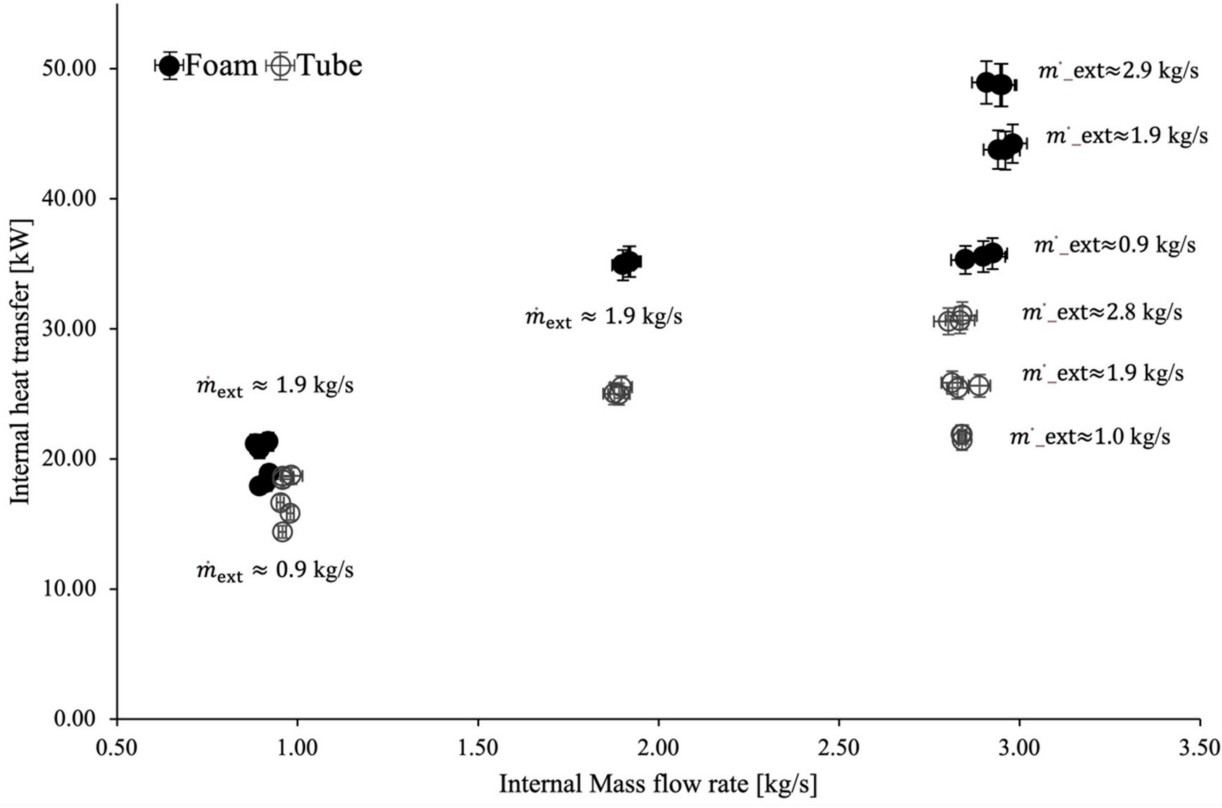

**Figure 4.** Internal heat transfer plotted versus the internal mass flow rate. (The external mass flow rates are shown next to each data set).

To quantify the heat transfer enhancement due to the ZA27 foam, the heat transfer increase was calculated for each mass flow rate combination according to $\frac{\dot{Q}_{\text{Foam}} - \dot{Q}_{\text{Tube}}}{\dot{Q}_{\text{Tube}}}$, using average values for the respective rates of heat transfer. The results are plotted in Figure 5 versus the internal mass flow rate. Heat transfer enhancement increases linearly with the internal mass flow rate. This can again be explained by the heat transfer $\dot{Q}_{\text{int}}$ between mass streams being limited by the inefficient heat transfer between the internal mass stream and the inner surface of the copper tube. In contrast, no systematic correlation between heat transfer enhancement and the external mass flow rate can be observed. For example, similar results are obtained for $\dot{m}_{\text{ext}} \approx 0.9, 1.9, 2.9$ kg/s as for $\dot{m}_{\text{int}} \approx 2.9$ kg/s. These data suggest othat the tube-filled foam heat exchanger should be operated at the lower external mass flow rates to minimize pressure drop across the foam material and decrease the required energy input to sustain this mass flow.

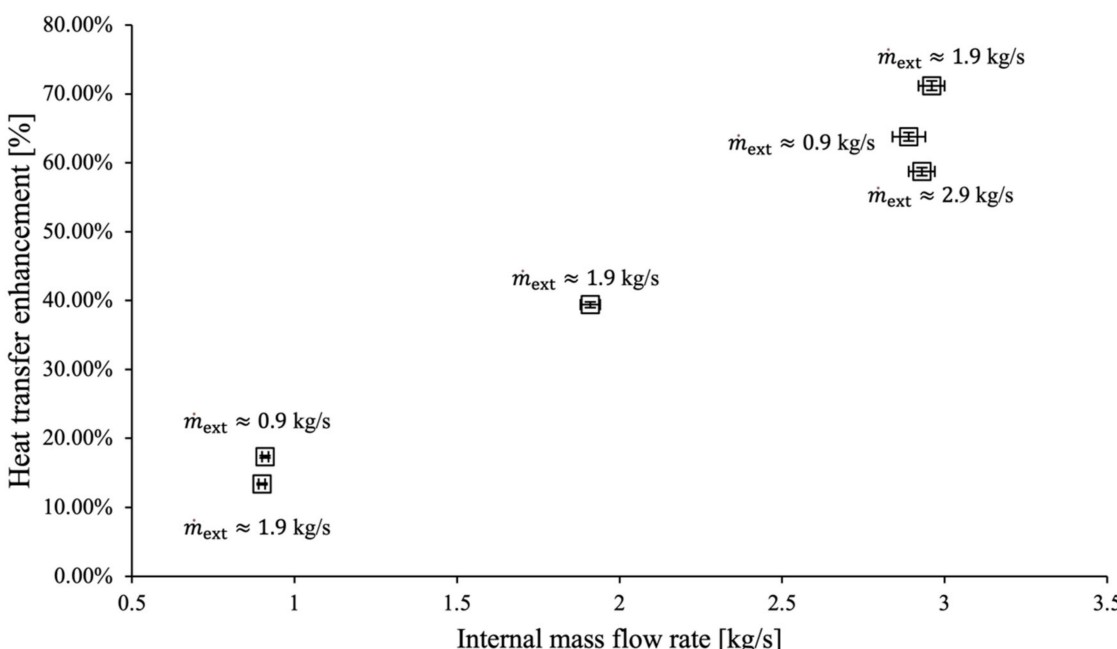

**Figure 5.** Heat transfer enhancement plotted versus the internal mass flow rate. (The external mass flow rates are shown next to each data set).

## 4. Conclusions

In this study, tube-filled foam heat exchangers were successfully manufactured using a single-step infiltration casting technique. As a result, direct bonding between an internal copper tube and surrounding ZA27 foam was achieved, minimizing thermal contact resistance. Testing the heat exchanger using water streams with mass flow rates between 0.9 kg/s and 2.9 kg/s and a temperature difference of approximately 42 K resulted in a heat transfer between mass streams of up to 49 kW. To quantify the heat transfer enhancement achieved by the metallic foam, testing was repeated using an identical setup, without the presence of ZA27 foam.

- Comparison of the testing data revealed a heat transfer enhancement of up to 71%.
- Overall performance was found to be limited by the inefficient heat transfer between the internal mass stream and the copper tube, due to a relatively small contact area.
- For future research, we will, therefore, consider compact heat exchangers containing heat-transfer-enhancing foams that are located both inside and outside a tube.
- In addition, ZA27 foam may be replaced with aluminium foam due to the higher thermal conductivity of this metal, and to prevent the formation of a zinc hydroxite layer (see Figure 1c) that may decrease heat transfer over time.

**Author Contributions:** Conceptualization, T.F.; methodology, T.F. and N.M.; validation, R.M. and N.M.; formal analysis, T.F.; investigation, R.M. and N.M.; resources, T.F.; data curation, R.M. and T.F.; writing—original draft preparation, T.F.; writing—review and editing, N.M. and T.F.; visualization, R.M. and T.F.; supervision, T.F.; project administration, T.F. All authors have read and agreed to the published version of the manuscript.

**Funding:** This research received no external funding.

**Institutional Review Board Statement:** Not applicable.

**Informed Consent Statement:** Not applicable.

**Data Availability Statement:** The data of this research can be obtained by contacting the corresponding author.

**Acknowledgments:** We would like to acknowledge the outstanding technical support by the University of Newcastle Mechanical Engineering workshop. In particular, the technical support by Mitch Gibbs was pivotal to the success of this research.

**Conflicts of Interest:** The authors declare no conflict of interest.

## Nomenclature

| | |
|---|---|
| $m$ | Mass |
| $E$ | Energy |
| $T_i$ | Local temperature |
| $h_i$ | Local enthalpy |
| $\dot{m}_{int}$ | Internal mass stream |
| $\dot{m}_{ext}$ | External mass stream |
| $\dot{Q}_{int}$ | Internal heat transfer rate |
| $\dot{Q}_{ext}$ | Dissipated thermal energy rate |
| $v$ | Velocity |
| $y$ | Elevation |
| $\dot{W}$ | Power |
| $\dot{Q}_{Foam}$ | Heat transfer rate of metal foam |
| $\dot{Q}_{Tube}$ | Heat transfer rate of metal tube |

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
