# Peer review of "Manufacturing and Characterization of Tube-Filled ZA27 Metal Foam Heat Exchangers"

_metals, doi:10.3390/met11081277_

Round 1
Reviewer 1 Report
REPORTS ON: metals-1295876 The aim proposed is reasonably interesting. However, there are number of weaknesses (mainly concerning to reproducibility) which induces to its REJECTION. The main reasons for this decision are below described:
1) Experimental details are rather and poorly described.
2) Reproducibility is absent. Error ranges of all involved physical elements are absent
3) Thermocouple details are absent. Error ranges are not indicated. Based on this attained results are speculated.
4) Error ranges are absent in Tables 1 and 2; and into Figs. 4 and 5
5) US and British English styles are used.
6) Chemical composition of proposed ZA alloy is rather and confused.
Author Response
We thank the reviewer for these valuable comments.
REPORTS ON: metals-1295876 The aim proposed is reasonably interesting. However, there are number of weaknesses (mainly concerning to reproducibility) which induces to its REJECTION. The main reasons for this decision are below described:
- Experimental details are rather and poorly described.
Additional experimental details have been added to the manuscript as briefly outlined in the subsequent points raised by the referee.
- Reproducibility is absent. Error ranges of all involved physical elements are absent
A short analysis of experimental uncertainty has been added to the manuscript. Error ranges have been included for all required sensors and relevant calculated values.
- Thermocouple details are absent. Error ranges are not indicated. Based on this attained results are speculated.
Information on the type and manufacturer of the thermocouples has been added: “All temperatures were measured using RS Pro IEC thermocouples type K with an accuracy of ±1.5% within the considered temperature range.”
- Error ranges are absent in Tables 1 and 2; and into Figs. 4 and 5
Error ranges have been added to the indicated Tables and Figures.
- US and British English styles are used.
The authors have attempted to adapt British spelling throughout the document.
- Chemical composition of proposed ZA alloy is rather and confused.
For improved clarity, the composition is now displayed in the new Table 1.
Reviewer 2 Report
Please find the attached file.

Author Response
In this paper, a new type of compact porous heat exchanger was proposed and its heat transferring capacity was investigated and compared its performance with standard tube model. The originality of this paper is high, but the following points should be added or revised in the paper.
We thank the reviewer for this assessment and the valuable comments.
- In Abstract, the authors comments “A significant heat transfer enhancement is observed”. But in Conclusion, the authors note “Comparison of the testing data revealed a heat transfer enhancement of up to 71%. However, overall performance was found to be limited by the inefficient heat transfer between the internal mass stream and the copper tube due to a relatively small contact area”. I recommend to revise the Abstract to show the effectiveness and limitation of the performance of the proposed porous tube qualitatively to match the Conclusion.
The Abstract has been reworded accordingly.
- How can the authors control the porous size, density and position for each particle in the porous tube?
The pore size is controlled via the particle size, which in turn is controlled by sieving the received particles. Previous measurements (e.g. reference [15]) have shown the particle bulk density to be uniform between multiple batches. Deviations of individual particle density are unlikely to affect the experiment, as particles are removed following casting. The particle position cannot be controlled; however, due to the spherical size and vibration during particle packing consistent bulk densities can be achieved.
- The authors mainly discuss the effects of mass flow late on the heat transferring performance, but not investigate the effects of porous tube geometries on it. The readers cannot understand how to design the porous tube in order to attain the appropriate heat transferring performance under the fixed flow rate condition. I recommend the authors to include the discussion of the effects of the porous tube geometries (e.g. relative density, inner and outer radius of the tube) on the performance.
This is an excellent idea but unfortunately beyond the scope of this investigation. In our future research, we indeed plan to investigate foam tubes with different pore sizes and macroscopic dimensions.
Reviewer 3 Report
Dear Authors,
Many thanks for your valuable work! It was really interesting more specifically when you talked directly about the novelties of your work! However, I wouldn't recommend the paper to be published at current format and must have major changes! Please find below comments on your work:
- The title must be more specific, you wrote it very general!
- The Abstract should be more descriptive and have a scientific structure but I couldnt see it in your paper!
- The introduction part and also your references are very few and outdated! You need to increase them to some reasonable amount! There are plenty works and also researchers all over the word that work on similar topics! In my opinion your introduction must be re written.
- Many items/terms are not mentioned in your manuscript! So, I recommend to add a nomenclature.
- I couldnt see you mention any error in your experiments or in your mathematical work. What is the accuracy of your model?
- All facilities in Figure 2 must be mentioned with their brands, capacity, error sources and ....
- Would be better to present a picture of your setup as well.
- Please highlight your major findings in your conclusions part with bullet! Would be easy to follow and make your research valuable!
Author Response
Dear Authors,
Many thanks for your valuable work! It was really interesting more specifically when you talked directly about the novelties of your work! However, I wouldn't recommend the paper to be published at current format and must have major changes! Please find below comments on your work:
We thank the reviewer for this assessment and the valuable comments.
The title must be more specific, you wrote it very general!
Following the recommendation of the reviewer, the title has been changed to “Manufacturing and Characterization of Tube Filled ZA27 Metal Foam Heat Exchangers”
The Abstract should be more descriptive and have a scientific structure but I couldnt see it in your paper!
The paper uses the official template the “Metals” journal and its recommended structure. The abstract now first states the overall purpose of the study, followed by the basic design of the study and finally the major findings as a result of the analysis. A summary of our conclusions has been added.
The introduction part and also your references are very few and outdated! You need to increase them to some reasonable amount! There are plenty works and also researchers all over the word that work on similar topics! In my opinion your introduction must be re written.
The introduction has been extended as recommended by the reviewer. Four recent publications have been included as the new references [10-13].
Many items/terms are not mentioned in your manuscript! So, I recommend to add a nomenclature.
As suggested, a nomenclature has been added to the manuscript.
I couldnt see you mention any error in your experiments or in your mathematical work. What is the accuracy of your model?
An analysis of experimental uncertainty has been added to the manuscript and included in the Tables and Figures.
All facilities in Figure 2 must be mentioned with their brands, capacity, error sources and ....
This information has been added for all sensors, i.e. the RS Pro IEC type K thermocouples and the RS 257-149 flowmeters, and considered in the analysis of experimental uncertainty.
Would be better to present a picture of your setup as well.
Following the suggestion of the reviewer, photographs have been added as new Fig. 2b)
Please highlight your major findings in your conclusions part with bullet! Would be easy to follow and make your research valuable!
The formatting has been adjusted as recommend by the reviewer.
Round 2
Reviewer 1 Report
The authors have made the revision accordi the reviewer's comments.
The article has reached an adequate level and can be considered for publication.
Author Response
We thank the referee for this assessment and the valuable input into further improving our manuscript.
Reviewer 3 Report
Dear Authors,
Thanks! many of my comments have been addressed carefully. Still, I believe the Abstract is not that appropriate needs qualitative data! It must provide a general descriptive of your research. You need to talk with numbers! You talked about increased! How much you increased!!
Author Response
We thank the referee for this positive assessment and the valuable input that has enabled us to further improve our manuscript.
Following the suggestion of the referee, key numerical data is now included in the abstract.